# Current Status and Future Prospects of Perinatal Stem Cells

**DOI:** 10.3390/genes12010006

**Published:** 2020-12-23

**Authors:** Paz de la Torre, Ana I. Flores

**Affiliations:** Grupo de Medicina Regenerativa, Instituto de Investigación Sanitaria Hospital 12 de Octubre (imas12), Avda. Cordoba s/n, 28041 Madrid, Spain; torre-merino.paz@h12o.es

**Keywords:** placenta, perinatal tissues, umbilical cord tissue/blood, chorion, decidua, amniotic fluid/membrane, regenerative medicine, mesenchymal stromal cells, nanomedicine, COVID-19

## Abstract

The placenta is a temporary organ that is discarded after birth and is one of the most promising sources of various cells and tissues for use in regenerative medicine and tissue engineering, both in experimental and clinical settings. The placenta has unique, intrinsic features because it plays many roles during gestation: it is formed by cells from two individuals (mother and fetus), contributes to the development and growth of an allogeneic fetus, and has two independent and interacting circulatory systems. Different stem and progenitor cell types can be isolated from the different perinatal tissues making them particularly interesting candidates for use in cell therapy and regenerative medicine. The primary source of perinatal stem cells is cord blood. Cord blood has been a well-known source of hematopoietic stem/progenitor cells since 1974. Biobanked cord blood has been used to treat different hematological and immunological disorders for over 30 years. Other perinatal tissues that are routinely discarded as medical waste contain non-hematopoietic cells with potential therapeutic value. Indeed, in advanced perinatal cell therapy trials, mesenchymal stromal cells are the most commonly used. Here, we review one by one the different perinatal tissues and the different perinatal stem cells isolated with their phenotypical characteristics and the preclinical uses of these cells in numerous pathologies. An overview of clinical applications of perinatal derived cells is also described with special emphasis on the clinical trials being carried out to treat COVID19 pneumonia. Furthermore, we describe the use of new technologies in the field of perinatal stem cells and the future directions and challenges of this fascinating and rapidly progressing field of perinatal cells and regenerative medicine.

## 1. Introduction

Cell therapy constitutes a strategy based on the use of cells as therapeutic agents. In this sense, studying and defining the type of cell to apply to a specific treatment is essential to success in regenerative medicine. This requires characterizing its safety and its ability to repair, replace or restore the biological function of damaged tissues and organs. The focal point of cell therapy is human cells, which can be somatic or stem cells. Stem cells are special human cells that can develop into many different cell types and simultaneously divide and self-renew to maintain their population. Depending on their origin they are classified as (1) embryonic stem cells (ESC), present in the inner cell mass of the blastocyst; (2) adult stem cells (ASC) that are present in numerous tissues of the human body including bone marrow, peripheral blood, and skin; and (3) perinatal stem cells, existing in the placenta and fetal annexes. ESC being pluripotent can give rise to cells of the three germ layers, while more specialized ASC play a role in replacing damaged/old cells in the tissues where they are present. Among the most fascinating sources for cell therapy are perinatal tissues, which include all the tissues from human term placentas and fetal annexes (i.e., amniotic and chorionic membranes, decidua, chorionic villi, chorionic plate, umbilical cord blood and tissue (Wharton’s jelly), and amniotic fluid). A significant advantage of the placenta is that it is a tissue discarded after birth with high yield cells that have been less exposed to infections reducing the possibility of disease transmission. The aim of this review is to give an overview of the interesting characteristics possessed by the stem and progenitor cell types isolated from the different perinatal tissues making them particularly interesting candidates for use in regenerative medicine. We review the preclinical and clinical studies conducted for a variety of indications, focusing on the pandemic from COVID-19, and the progress made in the field of nanotechnology related to perinatal stem cells. Our aim is to dissect the encouraging results obtained until now and the current and future challenges of these promising cell types.

## 2. The Placenta and Its Fetal Adnexa as a Source of Stem Cells

The placenta is a complex and temporary organ that forms the interface between the fetus and the mother responsible for fetal development and which ceases to function at week 40 of pregnancy. The two main components of the placenta, the fetal and the maternal, must interact efficiently to achieve a healthy pregnancy. The main functions of the placenta are to ensure the supply of nutrients to the fetus, remove metabolic products and prevent immune rejection to the conceptus. The placenta also has major endocrine functions and acts as a selective barrier protecting the fetus from maternal and environmental stressors, such as maternal hormones, xenobiotics, pathogens, and parasites.

Although umbilical cord blood has been used in transplants for over 30 years, the use of the placenta and its fetal annexes as a source of stem cells started around 10–15 years ago. The first international workshop on placenta-derived cells was organized in 2007, where the terminology, phenotype, and main properties of the cells isolated from different regions of placenta were defined [1]. Later, the International Placenta Stem Cell Society (IPLASS) was founded in September 2009 whose main purpose was to promote and advance research on placenta-derived stem cells, both in basic and clinical research, so that they could both contribute to benefit society. Placental stem cells or perinatal stem cells are derived from the placental blood or tissue. Among placenta-derived stem cells there are different types of cells, such as hematopoietic stem cells (HSC) derived from cord blood, epithelial stem cells, trophoblasts and mesenchymal stromal cells (MSC) derived from the placental tissues which include the amniotic and chorionic membranes, the amniotic fluid, the chorionic villi, the chorionic plate, the umbilical cord, and the decidua (Figure 1).

The most well-known perinatal cell types are perhaps the hematopoietic stem cells (HSC) from umbilical cord blood and mesenchymal stromal cells (MSC) isolated from umbilical cord blood and tissue, also known as Wharton’s jelly. The amniotic membrane that covers the placenta and the umbilical cord has a mixture of MSC and epithelial stem cells. Other parts of the placenta such as chorion membrane, and even amniotic fluid and the decidua, are all rich sources of stem and progenitor cells and we will refer to them collectively as perinatal cells [2].

### 2.1. Amniotic Fluid

Amniotic fluid (AF) contains stem cells that can be isolated and used in the future for clinical therapeutic purposes. AF is harvested in the second trimester of pregnancy, between the fifteenth and nineteenth week of gestation, during routine amniocentesis for prenatal diagnosis testing and the remaining sample is used for cell stem cell isolation (Figure 1). AF contains a heterogeneous cell population according to their morphologies and growth, in vitro biochemical characteristics and in vivo potential. AF mainly includes three types of cells: epithelioid (E) type cells derived from fetal skin and urine, amniotic fluid (AF) type derived from the fetal membranes and trophoblast, and fibroblastic (F) type cells derived from fibrous connective tissues and dermal fibroblasts [3], which vary proportionately in line with gestational age [4]. Based on plastic adherence, two populations of amniotic fluid cells can be isolated: the amniotic fluid mesenchymal stem cells (AFMSC) and the amniotic fluid stromal cells (AFSC).

AF-MSC are an unselected population of adherent cells isolated in serum-rich conditions from the second and third trimester AF which present characteristics of MSC. AF-MSC are plastic adherent cells following the minimal criteria of the first international workshop on placenta derived stem cells [1]. AF-MSC exhibit typical mesenchymal phenotype and are positive for MSCs markers CD90 and CD73, low levels of CD105, CD29, CD44 and HLA-ABC (MHC class I) but negative for CD34, CD45, CD 31, CD117, HLA-DR (MHC class II). AFMSC also express the pluripotency factor OCT4 and demonstrate a high proliferation capacity [5,6]. The differentiation potential of AFMSCs includes the mesodermal lines of adipocytes and osteocytes, as well as neuronal cells [5,7,8]. 

AFSCs are isolated by CD117 selection via either magnetic- or fluorescent-activated cell sorting (MACS or FACS, respectively) from the population of cells attached to the plastic. AFSCs have a phenotype that is between ESCs and adult MSCs reinforced by the expression of transcription factors of both pluripotency and mesenchymal cells [9]. AFSCs isolated from the second and third trimester human AF express c-Myc, Oct-4, and SSEA pluripotency-specific markers, but do not express Nanog, Klf4, SSEA3, Tra-1-60, Tra-1-81, or ALP [10]. Their mesenchymal phenotype is evident by the expression of CD29, CD44, CD73, CD90, CD105, as well as CxCR4, stromal cell-derived factor (SCF) 1 receptor, CD146, CD166, and CD184. AFSCs are positive for HLA class I (HLA-ABC) and negative for HLA class II (HLA-DR). AFSCs are a population of multipotent stem cells able to differentiate into mesoderm (bone, fat, cartilage, muscle, hematopoietic), endodermal (endothelial, hepatic) and ectodermal lineages (neuronal) [11,12,13,14,15]. Despite the intermediate phenotype between ESC and MSCs and that they are capable of forming embryoid bodies, AFSCs do not form teratomas when transplanted into immunocompromised mice [15,16]. AFSC have already demonstrated therapeutic potential for cardiovascular (ischemia-reperfusion injury, myocardial infarction), gastrointestinal (necrotizing enterocolitis), hematopoietic (congenital hematological diseases), musculo-skeletal (muscular dystrophy, regenerate bone in collagen alginate scaffolds), neurological (Krabbe globoid leukodystrophy, traumatic brain/ nerve injury, stroke, in utero treatment of spina bifida), respiratory (hyperoxia lung injury, lung hypoplasia) and urinary disorders (acute tubular necrosis, Alport syndrome) [9,17].

### 2.2. Amniotic Membrane

The amniotic membrane (AM) is the inner layer of the amniotic sac or extra-embryonic fetal membranes and is composed of three layers: an epithelial monolayer, an acellular basement layer, and a mesenchymal cell layer (Figure 1). AM is usually collected at term pregnancies after birth. AM includes two cell types, the amniotic membrane mesenchymal stromal cells (AMSC) and the amniotic epithelial cells (AEC) derived from the amniotic mesenchymal and the amniotic epithelial layers, respectively. After mechanical separation of the AM from the chorionic membrane, the AMSC and AEC are isolated by a two-step protocol. The tissue is first minced and digested with trypsin to remove AEC and is then digested with collagenase or a mixture of collagenase/DNase to obtain the AMSC [18,19].

The cell surface markers in AMSC are CD90, CD44, CD73, CD29, CD13, CD105, CD166, CD49e, CD10, and HLA-ABC. AMSC are also positive for the stem cells markers SSEA-3, SSEA-4 OCT-4, Rex-1, and GATA-4 [20]. Besides differentiating into the characteristic mesodermal lineages (osteogenic, chondrogenic, adipogenic), AMSC have the ability to differentiate into other cell types, such as neural and glial cells, skeletal muscle cells, cardiomyocytes, pancreatic and hepatic cells [21]. AM-MSCs have been used to treat lung fibrosis [22] and musculoskeletal disorders [23].

AEC are polygonal epithelial cells that express cytokeratin-7 and possess characteristics associated with a MSC phenotype such as the cell surface expression of CD90, CD105 and CD73, CD 117 and lack of CD45, CD34, CD14, CD79, and HLA-DR [24,25]. In addition, AEC also demonstrate pluripotent stem cell-like characteristics as they express cell surface antigens present in human embryonic stem cells such as SSEA-3 and -4, and TRA-1-60 and 1-81, and the expression of the transcription factors NANOG, SOX-2, and Oct-4 [26,27,28]. AEC are multipotent cells with the capacity to differentiate toward cells of the three germ layers [29]. AEC efficiently differentiate in vitro into osteocytes, adipocytes, cardiomyocytes, and myocytes (mesodermal), pancreatic and hepatic cells (endodermal), neural, and astrocytic cells (ectodermal). Besides their self-renewal ability and the expression of multipotency markers, AEC are safe because they do not form teratomas upon transplantation and are a readily available source of cells since they can be easily harvested from AM using non-invasive procedures [30]. In addition, AEC are an almost limitless source of stem cells with an average yield of more than 100 million AEC per discarded amnion that could be applied in cost-effective cellular therapies for the treatment of various conditions [29,31]. Similarly to AM grafts, AEC have also been widely investigated for their immune privilege [32]. There is no consensus on the mechanisms that mediate the low immunogenicity of AEC, although it is thought to be related to the expression of HLA class Ia antigens (HLA-A, -B, -C), and the unique HLA class Ib (e.g., HLA-G) cell surface molecules that are known to suppress immune responses. Likewise, it is thought to be related to the non-expression of HLA class II antigens (HLA-DR) neither to their co-stimulatory molecules [31]. The therapeutic effects of AEC have been studied in a broad variety of pathologies including ocular diseases [33], lung fibrosis [22], familial hypercholesterolaemia [34], cardiovascular pathologies [35], liver fibrosis [36], musculoskeletal disorders [37], and neurological diseases such as spinal cord injuries [38], Parkinson’s disease [39], traumatic brain injury [40] and multiple sclerosis [41]. 

Human AM is known to help the regeneration of damaged tissue. The application of intact human AM to heal skin wounds was reported for the first time more than a century ago. The human AM is a biocompatible scaffold with adequate mechanical properties, low immunogenicity, and anti-inflammatory, anti-microbial, and anti-fibrotic properties [42]. It has been explored for a variety of clinical applications such as skin wounds [43], endometrial fibrosis [44], reconstruction of the oral cavity [45], and ocular diseases [46], providing high biocompatibility after several months of implantation [47]. 

### 2.3. Chorionic Membrane

The chorionic membrane (CM) is the outer layer of the human extra-embryonic fetal membranes and connects the fetus to the maternal tissues (Figure 1). The CM is in close contact with the decidua and is separated from the amniotic membrane by a spongy layer of collagen fibers. The CM is composed of two layers: a mesenchymal layer and a trophoblastic layer. To isolate MSCs from the chorionic membrane (CMSC), the tissue is first digested by dispase to remove the trophoblastic layer, and later digested by collagenase or a mixture of collagenase/DNase [24,48]. CMSC are positive for CD90, CD73, CD105, CD166, CD29, CD13, CD54, and CD44 and negative for hematopoietic cell markers CD34, CD45, CD14, CD3, CD31, and HLA-DR [48,49]. CMSC also express the transcription factors OCT-4, GATA-2, STAT-3, and Notch-1 receptor [50]. CMSC have the multipotent capacity to differentiate into mesodermal (adipocytes, osteocytes), endodermal (pancreatic-like cells), and ectodermal (neuronal-like cells) lineage cells [51]. CMSC is a homogeneous cell population that has a smaller size than other perinatal cells offering some advantages on intravenous transplantation [49]. In recent years, CM has also been explored for use as a bioactive scaffold alone or together with AM in tissue engineering and regenerative medicine strategies for wound healing, burns, bone, and vascular diseases [52].

### 2.4. Chorionic Plate

The chorionic plate is made up of the amniochorionic membrane and the fetal vessels (Figure 1). The stem cells are isolated from the closest region to the umbilical cord once the amniotic membrane is removed and the isolated cells have a mesenchymal type phenotype [53]. Chorionic plate MSC (CP-MSC) are positive for CD73, CD90, CD105, CD44, and CD166, and negative for CD45, CD34, CD14, CD19, and HLA-DR CP-MSCs express CD106 and CD54 [53,54]. Chorionic plate MSC (CP-MSC) are of fetal origin and display superior proliferation, migration capacity, and immunomodulatory properties to MSC derived from other perinatal tissues such as umbilical cord (UC-MSC), chorionic villi (CV-MSC), and decidua (DMSC) [53,55]. In vitro differentiation assays show that CP-MSC are able to differentiate into adipogenic, osteogenic, chondrogenic, and hepatogenic lineages. The therapeutic effect of CP-MSC has been studied in hepatic diseases [56,57], neurological disorders such as optic nerve injury [58], and ovarian dysfunction [59].

### 2.5. Chorionic Villi

Chorionic villi (CV) are finger-like projections that sprout from the chorion, and together with the maternal tissue of the basal plate form the placenta (Figure 1). CV is a source of cells with a typical morphology and phenotype of multipotent mesenchymal stromal cells (CV-MSC) [25]. CV-MSC are fetal cells isolated through explant culture from chorionic fetal villi, but it has been reported that maternal contamination is quite possible [60,61]. CV-MSCs express the mesenchymal markers such as CD44, CD73, CD29, CD105, CD90, CD49e, CD166, and CD106, and HLA-ABC lack the expression of the hematopoietic markers CD45, CD34, AC133, CD19, and HLA-DR [62]. CV-MSC possess the non-immunogenic character of MSCs because they do not express the immune molecules CD14, CD56, CD80, CD83, or CD86. CV-derived cells possess multipotent properties, display high proliferation rate, and self-renewal capacity [62,63]. CV-MSC express the transcription factors H2.0-like Drosophila (HLX) and TGFB-induced factor (TGIF) that could be involved in their proliferation and differentiation capabilities [64]. CV-MSC differentiate toward adipocytes, osteocytes, chondrocytes, neurons, and hepatocyte lineage under appropriate induction conditions [62,65]. CV-MSC are genetically stable and express SOX2 but show no other pluripotency markers such as NANOG, OCT4 when the cells are isolated from placentas collected at term [62]. However, CV-MSC isolated from CV samples obtained between 11 and 13 weeks of gestation by villocentesis also express Oct-4, NANOG, and GATA4 [63]. CVMSC have been used to prevent endothelial dysfunction associated with diabetes and cardiovascular disease [66] in an in vitro model of breast cancer [67] and in cartilage tissue engineering [60]. 

### 2.6. Umbilical Cord 

The umbilical cord (UC) attaches the embryo to the placenta guaranteeing the continuous supply of nutrients and oxygen to the fetus during pregnancy (Figure 1). UC is composed of two umbilical arteries, one umbilical vein and a mucoid connective tissue surrounding the umbilical vessels (i.e., Wharton’s jelly). UC is an important source of both hematopoietic stem/progenitor cells (HSPC) and mesenchymal stromal cells (MSC). 

UC blood (UCB) is the blood that is present in the UC and the placenta after childbirth. The existence of HSPCs in UCB was demonstrated in the early 1970s [68], although it was not until the late 1980s that its clinical importance as a substitute to bone marrow for hematopoietic reconstruction was recognized and the first umbilical cord blood transplant was performed [69,70]. In 2018, the 30th anniversary of the first HSPCs transplant using UCB was celebrated [71]. UCB is a straightforward and more readily available source of HSPCs as it requires neither invasive harvesting nor additional cytokine treatments, as compared to other sources of HSPC such as bone marrow or mobilized peripheral blood. HSPC are multipotent cells that have self-renewal capacity and the ability to differentiate into all the different blood cell types (i.e., white blood cells, red blood cells, and platelets) that comprise the blood-forming system during the hematopoiesis process. HSPC from UCB have been widely used in clinical settings after collection and banking procedures for the treatment of severe hematological disorders, such as leukemia and Wiskott–Aldrich syndrome, and for regeneration of healthy blood cells after chemotherapy both in family-related and family-unrelated UC blood patients [72,73]. UCB-HSPC transplantation has several advantages over bone marrow or mobilized peripheral blood transplantation in terms of its ease of collection and availability for use off-the-shelf, more permissible donor HLA compatibility and lower severity of graft-versus-host disease. Despite these advantages, it is important to note that the amount of blood collected from a single UC is reduced and may be insufficient to provide the cell dose to treat adult patients, and the delayed neutrophil and platelet reconstruction increases the risk of mortality [74]. To overcome these limitations, several strategies are currently being used such as dual-cord blood transplantations [75], investigation into ex vivo expansion of cells from UCB [76], methods to improve the collection and standardization techniques [72,77], or the use of drugs or co-transfusion with MSC to improve the homing and engraftment of UCB cells [78,79]. 

Compared to HSPC from adult sources, UCB-HSPC are less mature and have longer telomeres and high telomerase activity, which confers a higher self-renewal capacity and higher proliferation potential [80]. Classically, HSPC are characterized by the expression of the CD34 antigen, but most of the CD34+ cells also express other antigens such as CD38, CD90, CD117, CD135, CD95, CD71, CD45RO, CD45RB, and AC133. The co-expression of these different antigens with CD34 antigen defines different HSPC populations, from more primitive or early progenitors to late progenitors and cells at an early stage of differentiation [81,82]. Besides its clinical use in the treatment of hematological disorders, cord blood has been studied as a treatment for several other pathologies such as liver and brain injury, stroke, hearing loss, diabetes, heart attack, and vision loss in both human clinical trials and preclinical studies [83,84,85,86,87,88].

In addition, the HSPC, MSC-like cells can be collected from UCB. Several groups have reported the isolation of MSCs from UCB although they are present at very low frequency and grow very slowly compared to the number of MSC isolated from other sources, such as bone marrow [89,90,91,92]. In fact, MSC are only successfully isolated from approximately 40% of UCB units [93]. These cells expressed CD13, CD29, CD44, CD146, CD73, CD105, CD166, and CD90, and had low expression of HLA-I but did not express CD14, CD31, CD34, CD45, CD51/61, CD64, CD106 and HLA-DR [91]. UC blood MSCs can be differentiated into osteocytes, chondrocytes, and adipocytes [91,92]. 

MSC are also isolated from Wharton´s Jelly (UC-MSC), the tissue surrounding the umbilical cord vessels by enzymatic digestion or explant methods. UC-MSC have a higher proliferation and self-renewal capacity due to a higher expression of telomerase activity but without developing tumorigenic formation after transplantation [94]. The expression of cell surface markers includes CD10, CD13, Cd29, CD44, CD73, CD90, CD105, and HLA-I, but do not express CD11, CD14, CD19, CD31, CD38, CD45, CD40, CD80, CD86, and HLA-II [95,96,97,98]. The expression of pluripotency markers such as the octamer-binding transcription factor (Oct-4), Nanog, sex-determining region Y box 2 (Sox2), Kruppel-like factor 4 (KLF-4), and stage specific embryonic antigen 4 (SSEA-4) suggests that UC-MSC are more primitive than cells from other adult sources [98,99]. UC-MSC are multipotent cells that can be differentiated toward cell types from all germ layers such as adipocytes, osteoblasts, chondrocytes, skeletal myocytes, cardiomyocytes, neuronal cells, hepatocyte, insulin-producing cells, endothelial cells, and germ-like cells [100,101,102,103,104,105,106,107,108,109]. UC-MSC have been extensively used in the treatment of numerous pathologies such as autoimmune diseases, immunologic post-transplant complications, lung injury, cardiovascular diseases, liver pathologies, musculoskeletal disorders, diabetes mellitus, and neurodegenerative disorders (reviewed in [110]).

### 2.7. Decidua

The decidua is the maternal component of placental tissues and is divided into three regions: the decidua basalis that originates at the site of embryo implantation, the decidua capsularis that encloses the embryo, and the decidua parietalis that covers the rest of the uterus and fuses with the decidua capsularis by the fourth month of pregnancy (Figure 1). Both decidua basalis and decidua parietalis are a source of MSCs [111,112]. Decidua-derived mesenchymal stromal cells (DMSC) presented similar size, morphology, phenotype, and mesodermal differentiation ability as other MSC but higher proliferation ability than bone marrow MSC [113] and stronger immunosuppressive potential than WJ-MSC [114]. DMSCs express CD44, CD90, CD105, CD117, CD73, CD29, CD13, CD146, and CD166 and HLA-ABC, but are negative for CD34, CD133, CD45, CD14, CD19, BCRP1, CD31, STRO-1, and the costimulatory molecules (CD40, CD80, CD83, and CD86), and HLA-DR 1 [111,112]. They express the pluripotency transcription factors Oct-4, Rex1, and the organogenesis regulator, GATA-4, but do not express SSEA-1, SSEA-4, TRA-1-60, and TRA-1-81 which suggests that DMSC are intermediate cells between embryonic and adult stem cells [111]. DMSC are multipotent cells that can be differentiated in vitro toward multiple cell types from all germ layers such as adipocytes, osteoblasts, chondrocytes, skeletal and cardiac myocytes, neuronal cells, hepatocytes and pulmonary cells [111,115,116]. DMSC are safe when injected intravenously at higher doses than those currently used in humans or even in repeated doses [113,117,118]. DMSC have been used to treat breast cancer affecting their growth and development [117], in multiple sclerosis modulating the clinical course decreased inflammatory infiltration of the central nervous system [118], in diabetes protecting endothelial cells from the toxic effects of high glucose [119], and in preeclampsia reducing inflammation, tissue damage and blood pressure [120].

## 3. Immunological Properties of Perinatal Stem Cells

The placenta plays an important role during pregnancy by modulating the maternal immune system and offering immunological protection to the fetus. Perinatal stem cells are not immunogenic and are in a state of immune tolerance. Perinatal stem cells do not express HLA class II antigens (HLA-DR) or the co-stimulatory molecules CD40, CD80, and CD86 that are required for T cell activation [121,122]. However, HLA-DR expression increases after in vitro stimulation with IFN-ɣ or when cultured without serum. In addition, perinatal stem cells express HLA-G, a non-classical MHC class I molecule, which is known to inhibit natural killer (NK) cells and CD8^+^T CD4^+^T cell proliferation [54]. HLA-G expression is also induced by IFN-γ on perinatal stem cells [123], although the precise role of IFN-γ on its immunomodulatory functions is still unclear.

Besides affecting the innate immune response, perinatal stem cells also affect the adaptive immune system and show potent immunosuppressive properties. Perinatal stem cells suppress the in vitro proliferation differentiation, and the function of immune cells such as T cells, dendritic cells (DC), and NK cells. This ability to suppress immune cells was observed in a cell–cell contact, in a trans-well system and using conditioned media suggesting that the immunomodulatory activity of perinatal stem cells is provided by a paracrine mechanism [118,124]. Several molecules are involved in the paracrine effect of perinatal stem cells which include the secretion of prostaglandin E2 indoleamine 2, 3-dioxygenase, NO, transforming growth factor-1, hepatocyte growth factor, and leukemia inhibitory factor, insulin like growth factor, and interleukin IL-10 [97,121]. The low immunogenicity and immunomodulatory properties of perinatal stem cells encourages their use in allogeneic clinical applications and in inflammatory and autoimmune diseases.

## 4. Biobanking of Perinatal Stem Cells and Tissues

UCB has been biobanked and used to treat patients for over 30 years. The first UCB transplantation was performed in France in 1988 in a child with Fanconi anemia. This first successful transplant gave way to the establishment and rapid expansion of UCB banks worldwide [73]. Both private and public cord blood banks have been developed in order to collect and cryopreserve UCB for both related and unrelated patients. UCB is collected and processed, the stem/progenitor cells are isolated and then the cryopreserved samples can be stored for over 20 years with efficient recovery of HSPCs [72,125,126]. By 2018, there were over 750.000 UCB units worldwide stored in public banks and almost 7.000.000 units stored in private banks according to the Parent’s Guide to Cord Blood Foundation [127], in a total of 533 banks worldwide (Table 1). 

Besides cord blood, various placental stem cells and tissues can be readily accessible for research and clinical purposes due to the advancements in their isolation and char-acterization techniques [128]. Umbilical cord tissue (Wharton Jelly), chorion membrane, decidua, amniotic fluid and amniotic membrane are sources of HSPC, MSC or AEC, all of which can be bio banked for future use for research and clinical purposes. In addition, perinatal tissues such as placental tissue and amniotic membrane can also be stored in biobanks. Currently, numerous biobanks that were only dedicated to umbilical cord blood storage have now introduced the storage of isolated MSC from umbilical cord tissue as an additional service. In addition, the amniotic membrane is also biobanked busing the patented AmnioCeptTM technology [129] to cryopreserve multiple AM samples from a single placenta, i.e., intact tissue and isolated cells, that could be used in several present and future therapeutic applications. Public biobanks will receive AM donations from placentas of babies born at term by elective caesarean section and in the absence of chorioamnionitis, chromosomal abnormalities or specific illnesses and lifestyle practices of the mothers [128]. Examples of public biobanks that offer the possibility to donate AM tissue under a specific authorization are the National Health Service Blood and Transplant in UK [130] or Donate Life America in USA [131].

The banking of perinatal-derived stem cells and tissues for future clinical use can be done either through public banks for allogeneic use or through private banks for autologous use [132]. For the biobanking processes it is important to emphasize the standardization of isolation and characterization procedures, and this requires specific protocols for high quality isolation, manipulation, cryopreservation and long-term storage for clinical distribution, or for future research investigations under current good manufacturing practice (GMP) conditions [133]. In order to preserve the efficacy from thawed cells and tissues it is important to control the composition of the cryoprotectant medium, the mode of freezing, as well as, the protocol for cell expansion before or after cryostorage [134,135]. 

## 5. Clinical Applications of Perinatal Stem Cells

At present, a number of clinical studies have been performed and there are also numerous clinical trials using perinatal-derived cells in a variety of diseases based on the benefits found in the use of these cells in preclinical models of human diseases [136]. The ClinicalTrials.gov (http://www.clinicaltrial.gov) and the EU Clinical Trials Register (http://www.clinicaltrialsregister.eu) are two of the twelve international trial registries where clinical trials of advanced cell therapies using perinatal cells can be found [137]. Although some of the completed trials have not yet published results, several others have demonstrated the safety and therapeutic benefits of perinatal stem cells. 

Due to the creation and expansion of UCB banks worldwide, over 40,000 umbilical cord blood transplantations have been performed to treat hematological and immunological diseases both in children and adult patients [138]. Interestingly, UCB has also been used to treat several other diseases such as cerebral palsy [139], autism [140], hypoxic ischemic encephalopathy [141], spinal cord injuries [142], stroke [143], diabetes [144], liver diseases or congenital cardiac defects [142]. A systematic review showed that the majority of these studies are in their early clinical stages and although some of them reported a therapeutic benefit, the lack of control groups in most studies significantly impairs the determination of efficacy [145]. More studies are necessary to confirm the methodology, standardize the treatments and the mode of reporting outcomes for a better understanding of clinical benefits and safety profile of the future use of UCB on those indications.

The use of cultured products from the UCB, such as UCB-MSC, and MSC from UC tissue is also being widely studied. A review of 281 clinical trials of advanced cell therapies using perinatal cells carried out during the decade between 2005 and 2015 showed that the most common cell source in these trials was cord blood and the most commonly used cells were the MSC and not HSPC [137]. Between 2007 and 2017, more than 170 clinical trials were registered [146], 155 of which are currently enrolling patients at 216 locations worldwide [147], and nearly 100 publications that employ UC-MSCs in numerous therapies [146,148]. Most of the studies using UC-MSC are in their early stages, i.e., clinical trials, case reports, or pilot studies, and the publications describe safe or positive outcomes although there is a lack of information about how the cells are isolated and formulated before administration [146]. UC-MSC have been used to treat a wide variety of medical conditions such as neurological, cardiovascular, hepatic, hematological and immunological, endocrine, pulmonary, ophthalmologic, musculoskeletal, and dermatologic pathologies (Table 2). An exhaustive revision of all these clinical studies using UC-MSC and published until August 2017 describes the number of cells per dose used, the number of doses, and the route of administration [148].

## 6. Clinical Use of Perinatal Stem Cells in the Treatment of COVID-19 Pneumonia 

The COVID-19 pandemic has become a huge challenge for health systems world-wide. It is a disease caused by the coronavirus SARS-CoV-2 (severe acute respiratory syndrome coronavirus 2) that has a high transmission rate and is associated with signif-icant fatality, particularly in risk groups. SARS-CoV-2 mainly affects the respiratory system, although it is a very complex disease in which other organs, such as kidneys, heart, nervous system, liver, gastrointestinal tract, and skin, can also be affected, and various pathophysiological mechanisms are also involved [186]. Most deaths are due to acute respiratory distress syndrome (ARDS) caused by an over activation of the immune system struggling to kill the virus, leading to a significant production of inflammatory factors resulting in severe cytokine storm [187]. High levels of inflammatory markers in blood which include C-reactive protein, ferritin, and D-dimers, and increased serum levels of several inflammatory cytokines and chemokines such as IL-6, TNF, GCSF, MCP-1 among others, have been associated with disease severity and death [188]. In addition to ARDS, cytokine storm contributes to secondary complications such as sepsis, hypercoagulability or fibrosis, thus, therapeutic interventions to control it are being tested. Steroid drugs such as dexamethasone and other corticosteroids capable of blocking immunological response seem useful in the short-term but dangerous in the long-term [189]. Likewise, targeted therapies to reduce the levels of individual cytokines have not offered the hoped for benefits [190]. Perinatal derived cells may represent an effective strategy to treat seriously ill COVID-19 patients, due to their immunomodulatory and regenerative potential and their ability to engraft into damaged tissues [191]. Several studies have reported the beneficial effects of MSC on different models of lung injury and fibrosis associated with a reduction of proinflammatory cytokines such as TNF and L-6, and an increase of anti-inflammatory cytokines such as IL-10 [192]. In addition, MSC release prostaglandin E2 (PGE2) and promote the reprogramming of macrophages toward a M2 phenotype which secrete anti-inflammatory cytokines, and play essential roles in angiogenesis, tissue maintenance, matrix remodeling, and repair [193,194]. The polarization of the macrophages may be essential for mitigation of the cytokine storm and resolution of the hyperinflammatory state in COVID-19 pneumonia. Besides inhibiting the overactivation of the immune system, MSC therapy may promote endogenous repair by modulating the lung microenvironment. MSC intravenously injected tend to accumulate in the lungs where they secrete numerous paracrine factors that play a relevant role in the protection and repair of lung tissue [195]. MSC acts by inhibiting apoptosis, limiting oxidative injury and enhancing regeneration [196].

Recently, the first clinical trial using UC-MSCs in the treatment of chronic obstructive pulmonary disease (COPD) has been published [197]. UC-MSCs transplantation significantly improved the quality of life and clinical conditions of COPD patients possibly due to the anti-inflammatory effects of UC-MSCs suggesting that their infusion could be used to treat COVID-19 pneumonia. As of 30 October, there was a total of 26 registered clinical trials using, or going to use perinatal-derived MSCs (Table 3). In most of these studies, the principal source of MSCs is UC tissue (23 out of 26), two of these studies use placenta-derived MSCs, and only one of the studies uses decidual stromal cells (DSC). Two of the studies have been completed and around 50% are still recruiting. The investigators of the one of the completed studies (NCT04288102, Phase 1/2) have already published the results of the previous phase 1 study conducted during the early stages of the COVID-19 outbreak [198]. Their results showed that intravenous infusion of UC-MSCs in COVID-19 patients was safe and well tolerated. An additional pilot study conducted to evaluate the efficacy of UC-MSCs for the treatment of severe COVID-19 showed an improvement in some of the clinical symptoms and a reduced lung inflammation with respect to the control group [199]. There are two additional studies exploring the use of acellular amniotic fluid (NCT04497389 and NCT04319731) in the treatment of patients hospitalized for COVID19-associated respiratory failure. In these studies the investigators hypothesize that the amniotic fluid without cells will reduce the inflammation in COVID-19 patients, and will possibly decrease the need for respiratory support.

## 7. Nanotechnology for Perinatal-Derived Stromal Cells

Nanotechnology used to treat diseases and prevent health issues is called Nanomedicine [200]. Linking nanotechnology to stem cell-based strategies can aid the replacement of injured or damaged tissues. In the field of stem cells and regenerative medicine, nanotechnology based approaches have been developed to control the differentiation process, to label and track transplanted cells, to improve the stem cell regenerative process and to facilitate drug delivery [201,202]. Several scaffolds based on hydrogels, nanofibers, nanotubes and nanoparticles (NPs) have been used to control stem cell proliferation and differentiation. 

Chorion-derived MSCs grown and differentiated over gold-coated collagen nanofibers (GCNFs) showed a significant increase in proliferation and a more advanced differentiated state for neuronal and cardiac differentiation, compared to control without substrate. The differentiation could be further accelerated by electrical stimulation due to the characteristics of these electrically conductive GCNFs [203]. Hydrogel matrix prepared by polymerizing carbon nanotubes into collagen type I supported the differentiation of human decidua parietalis stem cells into neural cells serving as a tool for future applications to obtain mature neurons at the site of injury [204]. AFC cultured in unmodified hydrogel-based scaffolds showed high levels of osteogenic differentiation, whereas AFC cultured over hydrogels coated with extracellular matrix (ECM)-derived oligopeptides maintained the pluripotency suggesting that interactions with ECM are important to support or inhibit the differentiation ability of ASC [205]. The treatment of bone loss and nonunion fractures is still a great challenge to achieve. UC-MSC showed increased osteogenesis and angiogenesis capacity in vivo when seeded in a recently developed nanocomposite scaffold made of a bioactive glass/gelatin mixture to treat critical size calvarial defects [206]. The combination of nanotechnology and perinatal-derived MSC could be a relevant and highly promising research field and provide significant contributions in the area of the musculoskeletal disorders.

The field of nanomedicine is now focusing on developing nanocarriers for targeted drug delivery combined with on-site drug release. Achieving targeted delivery of medication will improve efficacy and reduce side effects on non-target tissues. NPs have been widely investigated as carriers for targeted drug delivery to treat cancer. However, nanotechnology has not yet achieved a long tern stability of the nanoparticles, nor a specific localization in tumor tissues. Human MSC from the decidua of the human placenta (DMSC) have been observed migrating toward tumors in a preclinical model of breast cancer [117]. This characteristic makes them excellent cellular vehicles for drug-loaded nanoparticles that could also be modified to have a controlled release of the payload to avoid side effects and the premature death of the carrier cells [207,208]. Different strategies can also be used to load the nanoparticles with both, genes and drugs to improve therapeutic strategy and these nanoparticles and perinatal MSC can be used as transporters to release the therapeutic molecule on-demand [209,210].

For the use of perinatal cells in cell therapy applications there is a lack of reliable methods to monitor their biodistribution and pharmacokinetics once transplanted. There are only a few studies in which these parameters are investigated in perinatal cells, such as the use of fluorescent nanodiamonds to trace and quantify the biodistribution of MSC derived from the choriodecidual membrane of human placenta in miniature pigs [211]; or the use of polyethylene glycol-coated superparamagnetic iron oxide nanoparticles to label the placenta-MSC and to track their migration and distribution pattern into a clinically relevant glioblastoma mouse model [212].

In the long term, additional studies are necessary to provide insights into how research findings related to nanotechnology-based therapies can be applied to the use of perinatal derived stem cells. It is expected that new and exciting nanotechnology-perinatal stem cell platforms will arise.

## 8. Future Directions and New Prospects

The rapid advances in basic stem cell research using perinatal-derived cells and several reports of effective cell-based interventions in animal models of human disease have created high expectations for their use in regenerative medicine and cell therapies. There are many questions to be resolved to help the understanding of the role of perinatal stem cells in the human body and different therapeutic strategies have been used to treat diseased, injured, or aged tissues. However, there are many obstacles when translating in vitro science to the in vivo preclinical environment in order to take full advantage of their effect later in clinical settings [213]. 

Although some advances in the understanding of MSC biology have been made, several questions regarding the use of some of the cells derived from perinatal tissues have to be resolved before their clinical use. The most frequently perinatal-derived cells used in humans are UC-MSC, and MSC have different functional properties depending on how they are isolated, expanded, and administered [214]. A deeper understanding of the characteristics, properties, and function of the different stem cells derived from the perinatal tissues would help to decide which to use to treat each particular disease. 

It is also important to reach a better understanding of their paracrine mechanisms of action and their immunomodulatory properties. In addition, it is essential to determine the best culture conditions for their isolation and expansion under good manufacturing practices (GMP), the cell dose to use, and the regimen treatment for clinical approaches all requiring authorization from the European Medicines Agency (EMA) in Europe and the FDA in the United States. It is now well known that placental-derived stem cells exert their effects mostly due to paracrine mediators which act on endogenous cells to induce tissue repair and/or regeneration. Therefore, it is essential to identify which molecules are responsible for their therapeutic effects, as a cell-free treatment would be possible and would have several advantages such as safety concerns of cell transplantation and would also promote the differentiation of quiescent resident stem cells into the injured tissue. The therapeutic mechanisms of the different types of stem cells isolated from the placenta are poorly understood. 

Umbilical cord blood has been used and biobanked to treat hematological disorders for over 30 years. Recently, other perinatal stem cells have been used in patients in several indications including acute myocardial infarction, stroke, cancer, rheumatoid arthritis, and most recently to treat COVID-19 pneumonia with 26 registered clinical trials. Given that most clinical trials testing advanced perinatal cell therapies in humans are at an early stage, it is easy to think that this field is set to grow rapidly in the years to come. It is to be expected that UCB biobanks will also be important in the growing field of advanced perinatal cell therapies and will help to ensure their effectiveness and safety by having perinatal cells reach the clinic maintaining their properties and biological function.

The combination of nanotechnology and perinatal stem cell research is a new and highly promising field that could have a significant impact on human healthcare. However, the use of perinatal-derived MSCs in combination with nanoengineered devices and structures for cell therapy and tissue regeneration is still in its infancy, and more intensive in vivo and in vitro research is still required to be applied in humans. 

## Figures and Tables

**Figure 1 genes-12-00006-f001:**
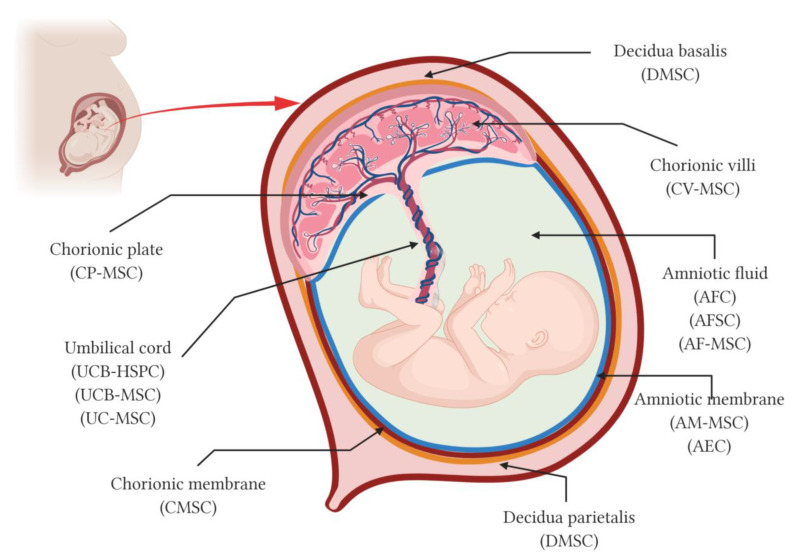
Schematic representation of perinatal tissues and perinatal stem cells. Anatomy of the human term placenta and its fetal annexes representing the main regions from which different types of perinatal stem cells have been isolated. AMSCs, amniotic membrane mesenchymal stromal cells; AEC, amniotic membrane epithelial cells; CMSCs, chorionic membrane mesenchymal stromal cells; CP-MSCs, chorionic plate mesenchymal stem cells; CV-MSCs, chorionic villi mesenchymal stromal cells; AFC, amniotic fluid cells; AFSC, amniotic fluid stem cells; AF-MSC, amniotic fluid mesenchymal stromal cells; UCB-HSPC, umbilical cord blood hematopoietic stem/progenitor cells; UCB-MSC, umbilical cord blood mesenchymal stromal cells; UC-MSCs, umbilical cord mesenchymal stromal cells; and DMSC, decidua-derived mesenchymal stromal cells. Created with BioRender.com.

**Table 1 genes-12-00006-t001:** Number of UC blood banks worldwide (according to the Parent’s Guide to Cord Blood Foundation).

	Public Banks	Family Banks
America	44	103
Europe	69	138
Asia	38	119
Africa	0	17
Oceania	3	2
**Total number of banks**	**154**	**379**

There are a total of 533 banks worldwide, of which, 379 correspond to private banks and 179 to public banks.

**Table 2 genes-12-00006-t002:** Summary of clinical applications of umbilical cord mesenchymal stromal cells.

Disorders	Disease Treated	Reference
Neurologic	Spinal cord injury	[149]
	Multiple Sclerosis	[150]
	Stroke	[151,152]
	Traumatic brain injury	[153]
	Amyotrophic lateral sclerosis	[154]
	Autism	[155]
	Cerebral palsy	[156]
	Hypoxic ischemic encephalopathy	[157]
	Vascular dementia	[158]
Cardiovascular	Acute myocardial infarction	[159]
	Systolic heart failure	[160]
Hepatic	Liver failure	[161,162]
	Transplant rejection	[163]
Hematologic	Graft versus host disease (acute, chronic)	[164,165]
	Leukemia	[166]
	Aplastic anemia	[167]
	Thrombocytopenia	[168]
Immunologic	Rheumatoid arthritis	[169]
	Ulcerative colitis	[170]
	Systemic lupus erythematosus	[171]
	HIV infection	[172]
Pulmonary	Severe idiopathic pulmonary fibrosis	[173]
	Bronchopulmonary dysplasia	[174]
Endocrine	Diabetes (Type I, Type II)	[175]
	Diabetic foot ulcer	[176]
Ophthalmologic	Retinitis pigmentosa	[177]
Musculoskeletal disorders	Becker muscular dystrophy	[178]
	Bone nonunion (fractured, infected)	[179,180]
	Duchenne muscular dystrophy	[181]
	Osteonecrosis of femoral head	[182]
	Cartilage regeneration	[183]
Dermatologic	Psoriasis vulgaris	[184]
	Cesarean scar defect	[185]

**Table 3 genes-12-00006-t003:** Clinical trials of perinatal-derived mesenchymal stromal cells registered in https://clinicaltrials.gov for treatment of COVID-19 as of 30th October 2020.

NCT Study	Status/No. Patients	Treatment	Study Type	Start Date	Location
NCT04366271	Recruiting/106	CT: UC-MSC Control: Standard care	Phase 2	7 May 2020	Spain
NCT04273646	Not yet recruiting/48	CT: UC-MSC Control: Placebo	N/A	20 April 2020	China
NCT04288102	Completed/100	CT: UC-MSC CG: Saline + HSA	Phase 1/2	5 March 2020	China
NCT04333368	Recruiting/40	CT: WJ-MSC Control: Saline	Phase 1/2	6 April 2020	France
NCT04494386	Recruiting/60	CT: UC-MSC Control: Placebo	Phase 1/2	23 July 2020	United States
NCT04490486	Not yet recruiting/21	CT: UC-MSC Control: Placebo	Phase 1	1 March 2021	United States
NCT04457609	Recruiting/40	CT: Standard care + UC-MSC Control: Standard care	Phase 1	7 July, 2020	Indonesia
NCT04355728	Active, not recruiting/24	CT: UC-MSC Control: Vehicle + Heparin	Phase 1/2	25 April 2020	United States
NCT04461925	Recruiting/40	CT: Standard care + P-MSC or UC-MSC Control: Standard care	Phase 1/2	2 May 2020	Ukraine
NCT04429763	Not yet recruiting/30	CT: UC-MSC Control: Placebo	Phase 2	July 2020	Colombia
NCT04293692	Withdrawn	CT: UC-MSC Control: Placebo	N/A	24 February 2020	China
NCT04452097	Not yet recruiting/9	CT: UC-MSC + Standard care	Phase 1	1 December 2020	United States
NCT03042143	Recruiting/75	CT: UC-MSC Control: Placebo	Phase 1/2	7 January 2019	United Kingdom
NCT04456361	Active, not recruiting/9	CT: WJ-MSC	Early Phase 1	16 April 2020	Mexico
NCT04269525	Recruiting/16	CT: UC-MSC	Phase 2	6 February 2020	China
NCT04565665	Recruiting/70	CT: UCB-MSC	Phase 1	29 July 2020	United States
NCT04437823	Recruiting/20	CT: UC-MSC + standard care Control: standard care	Phase 2	1 June 2020	Pakistan
NCT04371601	Active, not recruiting/60	CT: UC-MSC + standard care Control: Standard care	Early Phase 1	1 March 2020	China
NCT04573270	Completed/40	CT: UC-MSC Control: Placebo	Phase 1	24 April 2020	United States
NCT04313322	Recruiting/5	CT: WJ-MSC	Phase 1	16 March 2020	Jordan
NCT04390152	Not yet recruiting/40	CT: UC-MSC + standard care Control: Standard care	Phase 1/2	September 2020	Colombia
NCT04390139	Recruiting/30	CT: WJ-MSC Control: Placebo	Phase 1/2	13 May 2020	Spain
NCT04339660	Recruiting/30	CT: WJ-MSC Control: Saline	Phase 1/2	9 April 2020	China
NCT04389450	Recruiting/140	CT: PLX-PAD Control: Saline	Phase 2	1 October 2020	United States
NCT04451291	Not yet recruiting/20	CT: DSC	N/A	24 August 2020	Canada
NCT04614025	Recruiting/40	CT: PLX-PAD + standard care Control: Standard care	Phase 2	3 November 2020	Germany/ Israel

CT: cellular treatment; HSA: human serum albumin; UC-MSC: umbilical cord mesenchymal stromal cells; P-MSC: placenta mesenchymal stromal cells; UCB-MSC: umbilical cord blood mesenchymal stromal cells; WJ-MSC: Wharton Jelly mesenchymal stromal cells; ULSC: umbilical cord lining stem cells; PLX-PAD: allogeneic ex vivo expanded placental mesenchymal-like adherent stromal cells; DSC: decidual stromal cells.

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
