# Peer review of "Current Status and Future Prospects of Perinatal Stem Cells"

_genes, 2020, doi:10.3390/genes12010006_

Round 1

Reviewer 1 Report

The review is informative. However, there is high confusions about some definitions.

The umbilical cord blood / UC-HSC are not always considered as a perinatal tissue (or stem cell source) for all the community. So it lacks some information about that discutable point.

In all the document: Stromal is the right terminology (and not stem) for “UC-MSC (umbilical cord mesenchymal stem cells)" and ”UCB-MSC (umbilical cord blood mesenchymal stem cells)".

The terminology of the cells is not appropriately written in all the document included in the figure 1. For that please refer to the paper just accepted in Frontier special issue (Perinatal Derivatives and the Road to Clinical Translation: Part A): "Perinatal derivatives: where do we stand? A roadmap of the human placenta and consensus for tissue and cell nomenclature."
Antonietta R. Silini1*, Roberta Di Pietro2, Ingrid Lang3, Francesco Alviano4, Asmita Banerjee5, Mariangela Basile2, Veronika V. BorutinskaitÄ—6, Guenther EISSNER7, Alexandra Gellhaus8, Bernd Giebel9, Yong-Can Huang10, Aleksandar Janev11, Mateja Erdani Kreft11, Nadja Kupper12, Ana Clara Abadía Molina13, 14, Enrique G. Olivares14, 15, 13, Assunta Pandolfi16, Andrea Papait17, Michela Pozzobon18, Carmen Ruiz-Ruiz14, 13, Olga Soritau19, Sergiu Susman20, 21, Dariusz Szukiewicz22, Adelheid Weidinger5, Susanne Wolbank5, Berthold Huppertz12, ORNELLA PAROLINI17, 23

  • The right terminology for amniotic membrane cells is: AMSC (amniotic membrane mesenchymal stromal cells) and AEC (amniotic membrane epithelial cells).
  • The right terminology for chorionic membrane cells is: CMSC (chorionic mesenchymal stromal cells).
  • Lines 93 to 101: There is a confusion about the cells contained in the amniotic fluid: hAFC (epithelioid (E) type cells, which originate from fetal skin and urine; (2) amniotic fluid (AF) type cells which originate from the fetal membranes and trophoblast; and (3) F type cells that originate from fibrous connective tissues and dermal fibroblasts) + hAF-MSC + hAFSC. Please refer to the mentioned article in the special issue. Moreover "Stem" is not the right terminology for “AFMSCs, amniotic fluid mesenchymal stem cells”. MSC from the amniotic fluid (hAF-MSC) are plastic adherent cells defined following the minimal criteria of the consensus paper by Parolini et al. (Parolini et al. 2008, Spitzhorn et al. 2017).
  • Line 168 to 172: "Decellularized human AM has also been used as a biocompatible scaffold due to its anti-inflammatory, low antigen, feasibility, tolerance and low cost properties. The application of intact human AM was reported for the first time more than a century ago. It has been explored for a variety of clinical applications such as skin wounds [41] and endometrial fibrosis [42], providing 171 high biocompatibility after several months of implantation [43]" => It seems that “decellularized human AM” was reported for the first time more than a century ago. It has been explored for a variety of clinical applications such as skin wounds [41] and endometrial fibrosis [42], providing high biocompatibility after several months of implantation [43] => It is not right. Moreover What do you mean by: "low cost properties"? There is no citation.
  • Line 184-185: "CM-MSCs are less 184 investigated than other perinatal stem cells probably due to their reduced proliferative capacity". Do you have a citation ?
  • Line 324: there is no information about the biobanking of hAM (?)
  • Line 354: "Clinical applications of perinatal stem cells" => Not all clinical studies presented in this paper(*) are referred in the present review. (*) Review Cytotherapy. 2017 Dec;19(12):1351-1382.  doi: 10.1016/j.jcyt.2017.08.004. Epub 2017 Sep 28. Umbilical cord mesenchymal stromal cell transplantations: A systemic analysis of clinical trials. Alp Can  1 , Ferda Topal Celikkan  2 , Ozgur Cinar  2 => See for exemple musculoskeletal disorders in Table 2.

Figure 1:

The blue line needs to cover the chorion plate.

The terminology of the cells is wrongly written:

  • The right terminology for amniotic membrane cells is: AMSC (amniotic membrane mesenchymal stromal cells) and AEC (amniotic membrane epithelial cells).
  • The right terminology for chorionic membrane cells is: CMSC (chorionic mesenchymal stromal cells).
  • Amniotic fluid contains: hAFC, hAFSC and hAF-MSC. Stem is not the right terminology for “AFMSCs, amniotic fluid mesenchymal stem cells”. MSC from the amniotic fluid (hAF-MSC) are plastic adherent cells defined following the minimal criteria of the consensus paper by Parolini et al. (Parolini et al. 2008, Spitzhorn et al. 2017).
  • Stromal is the right terminology (and not stem) for “UC-MSC (umbilical cord mesenchymal stem cells)”.

Author Response

Dear Reviewer 1,

The authors gratefully acknowledge the comments made by the reviewer aimed to improve the quality of our work.

We have carefully revised our manuscript and addressed your comments. The replies to comments are given below. We have tried to respond to all your requests.

The umbilical cord blood / UC-HSC are not always considered as a perinatal tissue (or stem cell source) for all the community. So it lacks some information about that discutable point.

According to the reviewer comment we have added the following phase (lines 322-326):

The existence of HSPCs in UCB was demonstrated in the early 1970s [64], although it was not until the late 1980s that its clinical importance as a substitute to bone marrow for hematopoietic reconstruction was recognized and the first umbilical cord blood transplant was performed [65, 66]. In 2018 the 30th anniversary of the first HSPCs transplant using UCB was celebrated [67].

And we have also changed the abstract (Lines 22-25):

The primary source of perinatal stem cells is cord blood. Cord blood is a well-known source of hematopoietic stem/progenitor cells since 1974. Biobanked cord blood has been used to treat different hematological and immunological disorders for over 30 years. Other perinatal tissues that were routinely discarded as medical waste contain non-hematopoietic cells with potential therapeutic value.

In all the document: Stromal is the right terminology (and not stem) for “UC-MSC (umbilical cord mesenchymal stem cells)" and ”UCB-MSC (umbilical cord blood mesenchymal stem cells)".

Corrected on the Figure legend: UCB-MSCs, umbilical cord blood mesenchymal stromal cells UC-MSCs, umbilical cord mesenchymal stromal cells

Corrected on Table 3: UC-MSC: umbilical cord mesenchymal stromal cells; UCB-MSC: umbilical cord blood mesenchymal stromal cells

The terminology of the cells is not appropriately written in all the document included in the figure 1. For that please refer to the paper just accepted in Frontier special issue (Perinatal Derivatives and the Road to Clinical Translation: Part A): "Perinatal derivatives: where do we stand? A roadmap of the human placenta and consensus for tissue and cell nomenclature."
Antonietta R. Silini1*, Roberta Di Pietro2, Ingrid Lang3, Francesco Alviano4, Asmita Banerjee5, Mariangela Basile2, Veronika V. BorutinskaitÄ—6, Guenther EISSNER7, Alexandra Gellhaus8, Bernd Giebel9, Yong-Can Huang10, Aleksandar Janev11, Mateja Erdani Kreft11, Nadja Kupper12, Ana Clara Abadía Molina13, 14, Enrique G. Olivares14, 15, 13, Assunta Pandolfi16, Andrea Papait17, Michela Pozzobon18, Carmen Ruiz-Ruiz14, 13, Olga Soritau19, Sergiu Susman20, 21, Dariusz Szukiewicz22, Adelheid Weidinger5, Susanne Wolbank5, Berthold Huppertz12, ORNELLA PAROLINI17, 23

  • The right terminology for amniotic membrane cells is: AMSC (amniotic membrane mesenchymal stromal cells) and AEC (amniotic membrane epithelial cells).
  • The right terminology for chorionic membrane cells is: CMSC (chorionic mesenchymal stromal cells).

According to the reviewer comment we have follow the proposed consensus nomenclature in the article by Salini et al. 2020.

AMSC was applied to amniotic membrane mesenchymal stromal cells in Figure 1, Figure legend and point 2.2 Amniotic membrane

AEC was applied to amniotic membrane epithelial cells in Figure 1, Figure legend and point 2.2 Amniotic membrane

CMSC was applied to chorionic mesenchymal stromal cells in Figure 1, Figure legend and point 2.3 Chorionic membrane

  • Lines 93 to 101:There is a confusion about the cells contained in the amniotic fluid: hAFC (epithelioid (E) type cells, which originate from fetal skin and urine; (2) amniotic fluid (AF) type cells which originate from the fetal membranes and trophoblast; and (3) F type cells that originate from fibrous connective tissues and dermal fibroblasts) + hAF-MSC + hAFSC. Please refer to the mentioned article in the special issue. Moreover "Stem" is not the right terminology for “AFMSCs, amniotic fluid mesenchymal stem cells”. MSC from the amniotic fluid (hAF-MSC) are plastic adherent cells defined following the minimal criteria of the consensus paper by Parolini et al. (Parolini et al. 2008, Spitzhorn et al. 2017).

According to the reviewer comment:

1) We have corrected the previous phrase about the cells contained in the amniotic fluid and added the following phrase (lines (110-112) and we refer to the mentioned article suggested by the reviewer:

AF mainly includes three types of cells: epithelioid (E) type cells derived from fetal skin and urine, amniotic fluid (AF) type derived from the fetal membranes and trophoblast, and fibroblastic (F) type cells derived from fibrous connective tissues and dermal fibroblasts

2) The term Stem has been corrected for AF-MSCs, amniotic fluid mesenchymal stem cells to AF-MSCs, amniotic fluid mesenchymal stromal cells.

3 ) We have included the following phrase (lines 117-118): AFMSC are plastic adherent cells following the minimal criteria of the first international Workshop on Placenta Derived Stem Cells.

Line 168 to 172: "Decellularized human AM has also been used as a biocompatible scaffold due to its anti-inflammatory, low antigen, feasibility, tolerance and low cost properties. The application of intact human AM was reported for the first time more than a century ago. It has been explored for a variety of clinical applications such as skin wounds [41] and endometrial fibrosis [42], providing 171 high biocompatibility after several months of implantation [43]" => It seems that “decellularized human AM” was reported for the first time more than a century ago. It has been explored for a variety of clinical applications such as skin wounds [41] and endometrial fibrosis [42], providing high biocompatibility after several months of implantation [43] => It is not right. Moreover What do you mean by: "low cost properties"? There is no citation.

The paragraph has been rewritten (lines 225-231):

Human AM is known to help the regeneration of damaged tissue. The application of intact human AM to heal skin wounds was reported for the first time more than a century ago. The human AM is a biocompatible scaffold with adequate mechanical properties, low immunogenicity, and anti-inflammatory, anti-microbial, and anti-fibrotic properties [42]. It has been explored for a variety of clinical applications such as skin wounds [43], endometrial fibrosis [44], reconstruction of the oral cavity [45], and ocular diseases [46], providing high biocompatibility after several months of implantation [47].

  • Line 184-185: "CM-MSCs are less 184 investigated than other perinatal stem cells probably due to their reduced proliferative capacity". Do you have a citation ?

Eliminated.

  • Line 324: there is no information about the biobanking of hAM (?)

The authors gratefully acknowledge the comments made by the reviewer and we have added the following pharse (lines 443-447):

In addition, the amniotic membrane is also biobanked busing the patented AmnioCeptTM technology (https://parentsguidecordblood.org/en/news/amniotic-membrane-placenta-part-2) to cryopreserve multiple AM samples from a single placenta, i.e., intact tissue and isolated cells, that could be used in several present and future therapeutic applications.

  • Line 354:"Clinical applications of perinatal stem cells" => Not all clinical studies presented in this paper(*) are referred in the present review. (*) Review Cytotherapy. 2017 Dec;19(12):1351-1382.  doi: 10.1016/j.jcyt.2017.08.004. Epub 2017 Sep 28. Umbilical cord mesenchymal stromal cell transplantations: A systemic analysis of clinical trials. Alp Can  1 , Ferda Topal Celikkan  2 , Ozgur Cinar  2 => See for exemple musculoskeletal disorders in Table 2.

The reviewer is right that not all clinical studies are presented in this paper. We have already mentioned this fact in lines 491-493 and cited the review that he/she said. We had previously written:

An exhaustive revision of all these clinical studies using UC-MSC and published until August 2017 describes the number of cells per dose used, the number of doses, and the route of administration [140].

We have included some of the studies included in Can et al. to create our table, but we have also included some others that were not included in their study. For example, in musculoskeletal disorders, we have added the use of the UC-MSC for cartilage regeneration that is a very recent published study and was not included in the previous study:

Song, J.S., et al., Implantation of allogenic umbilical cord blood-derived mesenchymal stem cells improves knee osteoarthritis outcomes: Two-year follow-up. Regen Ther, 2020. 14: p. 32-39.

To avoid any confusion about this matter when reading the table we have changed its header to:

Table 2. Summary of clinical applications of umbilical cord-mesenchymal stromal cells

Figure 1:

The blue line needs to cover the chorion plate. Corrected in Figure 1

The terminology of the cells is wrongly written:

  • The right terminology for amniotic membrane cells is: AMSC (amniotic membrane mesenchymal stromal cells) and AEC (amniotic membrane epithelial cells). Corrected in Figure 1 and point 2.2 Amniotic membrane
  • The right terminology for chorionic membrane cells is: CMSC (chorionic mesenchymal stromal cells). Corrected in Figure 1 and in point 2.3 Chorionic membrane
  • Amniotic fluid contains: hAFC, hAFSC and hAF-MSC. Stem is not the right terminology for “AFMSCs, amniotic fluid mesenchymal stem cells”. MSC from the amniotic fluid (hAF-MSC) are plastic adherent cells defined following the minimal criteria of the consensus paper by Parolini et al. (Parolini et al. 2008, Spitzhorn et al. 2017). Corrected in Figure 1and in point 2.1 Amniotic fluid
  • Stromal is the right terminology (and not stem) for “UC-MSC (umbilical cord mesenchymal stem cells)”. Corrected in Figure 1

Reviewer 2 Report

The paper of de la Torre et al. is dealing with the current and future use of placental stem cells subpopulations. The review give many descriptive information about placental stem cells but these are all already available in medical literature. The authors hypothesize a possible use in the patient affected by covid-19 but no mechanism of action is suggested. The paragraph dealing with nanotechnology for perinatal-derived stromal cell is out of topic.

Author Response

Dear Reviewer 2

We have carefully revised our manuscript and addressed your comments. The replies to comments are given below. We have tried to respond to all your requests.

The review give many descriptive information about placental stem cells but these are all already available in medical literature.

The reviewer is right that there is already available medical literature describing the different perinatal stem cells, however, the most available literature offers separately treated overviews. We have not found any review compiling an overall description of all types of cells derived from the placenta, including the hematopoietic stem cells obtained from the umbilical cord blood. Besides, we have included here the topic of bio banking, both, for the well-known collection of umbilical cord blood, and for the not so well-known banking of other types of perinatal cells. We have also made an overview of present clinical uses of perinatal cells, and focus on the clinical trials on COVID-19 pneumonia making a compile which is not already available in medical literature. Finally, we describe the use of new technologies and the future directions and challenges in the field of perinatal stem cells.

The authors hypothesize a possible use in the patient affected by covid-19 but no mechanism of action is suggested.

We gratefully acknowledge this comment and we have included the suggested mechanisms of action of perinatal stromal cells (lines 500-529):

The COVID-19 pandemic has become a huge challenge for health systems worldwide. It is a disease caused by the coronavirus SARS-CoV-2 (severe acute respiratory syndrome coronavirus 2) that has a high transmission rate and is associated with a significant fatality, particularly in risk groups. SARS-CoV-2 mainly affects the respiratory system, although it is a very complex disease in which other organs, such as the kidneys, heart, nervous system, liver, gastrointestinal tract, and skin, can also be affected, and various pathophysiological mechanisms are also involved [181]. Most deaths are due to acute respiratory distress syndrome (ARDS) caused by an over activation of the immune system struggling to kill the virus, leading to a large production of inflammatory factors resulting in severe cytokine storm [182]. High levels of inflammatory markers in blood which include C-reactive protein, ferritin, and D-dimers, and increased serum levels of several inflammatory cytokines and chemokines such as IL-6, TNF, GCSF, MCP-1 among others, have been associated with disease severity and death [183]. In addition to ARDS, cytokine storm contributes to secondary complications such as sepsis, hypercoagulability or fibrosis, thus, therapeutic interventions to control it are being tested. Steroid drugs such as dexamethasone and other corticosteroids capable of blocking immunological response seem useful in the short-term but dangerous at long-term [184]. Likewise, targeted therapies to thumping down the levels of individual cytokines have not offered the hoped benefits [185]. Perinatal derived cells may represent an effective strategy to treat seriously ill COVID-19 patients, due to their immunomodulatory and regenerative potential and theit ability to engraft into damaged tissues [186]. Several studies have reported the beneficial effects of MSC on different models of lung injury and fibrosis associated to a reduction of proinflammatory cytokines such as TNF and L-6, and an increase of anti-inflammatory cytokines such as IL-10 [187]. In addition, MSC release prostaglandin E2 (PGE2) and promote the reprogramming of macrophages toward a M2 phenotype which secrete anti-inflammatory cytokines, and play essential roles in angiogenesis, tissue maintenance, matrix remodeling, and repair [188, 189]. The polarization of the macrophages may be essential for mitigation of the cytokine storm and resolution of the hyperinflammatory state in COVID-19 pneumonia. Besides inhibiting the overactivation of the immune system, MSC therapy may promote endogenous repair by modulating the lung microenvironment. MSC intravenously injected tend to accumulate in the lungs where they secrete numerous paracrine factors that play a relevant role in the protection and repair of lung tissue [190]. MSC acts inhibiting apoptosis, limiting oxidative injury and enhancing regeneration [191].

The paragraph dealing with nanotechnology for perinatal-derived stromal cell is out of topic.

We accept the reviewer's comment but consider that the use of perinatal derived cells and nanotechnology is an important topic to include because we have tried here to give an overview of what has been done so far and the potential future use of this type of cells. The use of nanotechnology to monitor perinatal cells once transplanted, or to use them as carriers of nanodrugs to avoid important side effects of traditional therapeutic drugs is a very important field that is under the subject of this review: “current status and future prospects”.

We have also sent the article to re-review again by an English native speaker.

Therefore, we ask the reviewer to reconsider his/her decision.

Round 2

Reviewer 1 Report

Dear authors 
I would great appreciate a letter to the reviewer explaining :

  • why you consider umbilical cord blood and derived cells as perinatal tissue/cells. You have to introduce this literature/research debate.
  • if you implemented the denomination from the Frontier review I suggested. Doing that, you will be able to realize for example that in your figure, the blue line (amnion) has to cover also the umbilical cord. 
  • why you did not revised the clinical trials table 2.

Thank by advance 

Author Response

I would great appreciate a letter to the reviewer explaining :

  • why you consider umbilical cord blood and derived cells as perinatal tissue/cells. You have to introduce this literature/research debate.

We are sorry, but certainly we have not been able to understand what the problem is by considering umbilical cord blood as a perinatal tissue and a source of stem cells. The literature is full of references that show this idea (see below) and, even companies like Merck in its promotional information about stem cells present cord blood cells as prototype of perinatal stem cell. Perinatal refers to the period that begins at 22 completed weeks of gestation and ends seven full days after birth. And blood is a tissue given that it is a collection of similar specialized cells that serve particular functions. The umbilical cord, together with the placenta and amniotic fluid, are the perinatal sources of stem cells. Of the tissues contained in the cord, blood is known to contain both hematopoietic stem cells and multipotent mesenchymal stromal cells (Bieback K, Kern S, Kluter H, Eichler H. Critical parameters for the isolation of mesenchymal stem cells from umbilical cord blood. Stem Cells 2004; 22: 625–34). According to these evidences, we have not found any reason to introduce the debate that the reviewer invites us to hold.

  • Abbaspanah, B., Momeni, M., Ebrahimi, M., & Mousavi, S. H. (2018). Advances in perinatal stem cells research: a precious cell source for clinical applications. Regenerative medicine13(05), 595-610.
  • StefaÅ„ska, K., Bryl, R., Hutchings, G., Shibli, J. A., & Dyszkiewicz-KonwiÅ„ska, M. (2020). Human umbilical cord stem cells–the discovery, history and possible application. Medical Journal of Cell Biology8(2), 78-82.
  • Brown, K. S., Rao, M. S., & Brown, H. L. (2019). The future state of newborn stem cell banking. Journal of clinical medicine8(1), 117.
  • Deus, I. A., Mano, J. F., & Custódio, C. A. (2020). Perinatal tissues and cells in tissue engineering and regenerative medicine. Acta Biomaterialia.
  • Richard L. Haspel, Karen K. Ballen, Cord Blood Transplantation: Therapeutic Use of Perinatal Stem Cells. Perinatal Stem Cells (Research and therapy), Academic Press, 2018 Pages 239-248,ISBN 9780128120156,Editor(s): Anthony Atala, Kyle J. Cetrulo, Rouzbeh R. Taghizadeh, Sean V. Murphy, Curtis L. Cetrulo,

  • if you implemented the denomination from the Frontier review I suggested. Doing that, you will be able to realize for example that in your figure, the blue line (amnion) has to cover also the umbilical cord. 

We have implemented the denomination from the Frontier review as the reviewer suggested. Because of that, and according to the upper part of Figure 1 of the Frontier review, we did not cover the umbilical cord with the amnion as the authors of the Frontier´s did. Our figure is a general overview of the different parts of the placental tissues and the cells obtained, and not a detailed description as you can see in the lower part Figure 1 of the Frontier review.

  • why you did not revised the clinical trials table 2.

We do not understand what the reviewer means about why we did not revise the clinical trials in our Table 2.

We already responded about this matter, but we will respond it again.

To create our Table 2, we have included some (but not all) of the clinical studies of the previous published paper that the reviewer mentioned (Can et al., 2017) for each one of the pathologies, and not only for musculoskeletal disorders. In addition, we have also included some additional studies because that article is from 2017.

We have already mentioned that in the first version of the article (lines 423-425):

An exhaustive revision of all these clinical studies using UC-MSC and published until August 2017 describes the number of cells per dose used, the number of doses, and the route of administration [140].

  • the suggestion I did regarding the biobanking of hAM is notappropriately addressed. What about the national banking in public banks?

We have added the following phrase to address the national baking in public banks hoping to appropriately address this matter as recommended (lines 373-380):

Public biobanks will receive AM donations from placentas of babies born at term by elective caesarean section and in the absence of chorioamnionitis, chromosomal abnormalities or specific illnesses and lifestyle practices of the mothers [127]. Examples of public biobanks that offer the possibility to donate AM tissue under a specific authorization are the National Health Service Blood and Transplant in UK (https://www.nhsbt.nhs.uk/what-we-do/transplantation-services/tissue-and-eye-services/tissue-donation/become-a-donor/living-amniotic-membraneplacenta-donation-programme/) or Donate Life America in USA (https://www.donatelife.net/types-of-donation/birth-tissue/).

Reviewer 2 Report

Tha paper much improved.

Author Response

Thank you very much for your positive comment.
